# Chemical and Biocatalytic Routes to Arbutin [note 1]

**DOI:** 10.3390/molecules24183303

**Published:** 2019-09-11

**Authors:** Hangyu Zhou, Jing Zhao, Aitao Li, Manfred T. Reetz

**Affiliations:** 1State Key Laboratory of Biocatalysis and Enzyme Engineering, Hubei Collaborative Innovation Center for Green Transformation of Bio-Resources, Hubei Key Laboratory of Industrial Biotechnology, School of Life Sciences, Hubei University, Wuhan 430062, China; 201711110710872@stu.hubu.edu.cn (H.Z.); zhaojing@hubu.edu.cn (J.Z.); 2Max-Planck-Institut für Kohlenforschung, Kaiser-Wilhelm-Platz 1, 45470 Mülheim, Germany; 3Chemistry Department, Philipps-University, Hans-Meerwein-Str. 4, 35032 Marburg, Germany

**Keywords:** arbutin, glycosyltransferases, glucosides, shikimate pathway, cosmetics, directed evolution, Chinese folk medicines

## Abstract

Arbutin (also called β-arbutin) is a natural product occurring in the leaves of a variety of different plants, the bearberries of the *Ericaceae* and *Saxifragaceae* families being prominent examples. It is a β-glucoside derived from hydroquinone (HQ; 1,4-dihydroxybenzene). Arbutin has been identified in traditional Chinese folk medicines as having, inter alia, anti-microbial, anti-oxidant, and anti-inflammatory properties that useful in the treatment of different ailments including urinary diseases. Today, it is also used worldwide for the treatment of skin ailments by way of depigmenting, which means that arbutin is a component of many products in the cosmetics and healthcare industries. It is also relevant in the food industry. Hundreds of publications have appeared describing the isolation, structure determination, toxicology, synthesis, and biological properties of arbutin as well as the molecular mechanism of melanogenesis (tyrosinase inhibition). This review covers the most important aspects with special emphasis on the chemical and biocatalytic methods for the production of arbutin.

## 1. Introduction

The natural product arbutin is the β-d-glucopyranoside of hydroquinone (HQ; 1,4-dihydroxybenzene) (Figure 1A), which occurs in the leaves of a number of medicinal plants, for example, in “bearberries” of the *Ericaceae* and *Saxifragaceae* families [1,2,3,4,5,6]. (Figure 1B).

The leaves of these plants have been used traditionally for hundreds/thousands of years as folk medicines by the native inhabitants of China and the American continents (e.g., Cherokee Indians), especially for wound healing and the treatment of urinary tract infections [1,2,3,4,5,6]. Therapeutic applications of arbutin are still relevant today [7,8,9,10,11,12], not just in China, where it is applied inter alia in the treatment of asthma [13]. Due to its (mildly) anti-microbial, anti-oxidant, and anti-inflammatory activity, it is also widely used in the cosmetic and healthcare industries worldwide, and is also of relevance in the food industry [1,2,3,4,5,6,7,8,9,10,11,12]. For example, since arbutin inhibits the melanogenesis process by the inhibition of tyrosinase, it is used as a depigmenting agent on skin, preventing and eliminating the growth of dark spots [1,2,3,4,5,6,7,8,9,10,11,12,13,14,15]. A possible mechanism of tyrosinase-inhibition by arbutin has been proposed on the basis of molecular dynamics (MD) computations, leading to the identification of crucial protein-arbutin interactions (Figure 2), which are different from those exhibited by other inhibitors [11]. Relevant to this issue is another study concerning the discrimination between alternative substrates and inhibitors of tyrosinase [12].

It has also been found that the really active component of arbutin is actually HQ, which is formed in small amounts by (enzymatic) hydrolysis on or in the skin, and that when HQ is used alone, toxic effects may arise [16]. This is probably due to tyrosinase-mediated HQ-oxidation [3,17]. It has been reported that arbutin inhibits tyrosinase-activity without influencing its biosynthesis in human melanocyte cultures [18,19]. Recent studies on biological effects of arbutin have focused on the reassessment of its antioxidant activity [15] and its function as an attenuator of LPS-induced lung injury via the Sirt1/Nrf2/NF-kBp65 pathway [20].

Historically, arbutin was first isolated by A. Kawalier in 1852 [21], and its rough composition was determined in 1858 by A. Strecker, who identified hydrolysis glucose and HQ [22]. C. Mannich was the first to obtain pure arbutin by synthesis and careful purification [23] (see Section 2). Its precise structure was determined during the last half of the 20th century, culminating in the recent complete characterization on the basis of Nuclear Magnetic Resonance-, Infrared- and UV/Vis-spectroscopic measurements as well as circular dichroism (CD) data and x-ray structure determination [24,25]. Arbutin is sometimes called β-arbutin to distinguish it from the diastereomeric form α-arbutin (α-configuration at the anomeric center), which is not a natural product. Syntheses of α-arbutin have been reported, and it has similar, but not identical biological properties [1,2,3,4,5,6,26,27,28]. In this review, the β-arbutin designation is not used, and we refer to compound **1** (Figure 1A) simply as arbutin.

The synthesis of arbutin is not particularly challenging. Retrosynthetic analysis suggests the coupling of two components, HQ and glucose. Any synthesis using man-made reagents and/or catalysts can be expected to require a routine multi-step process where the protection and deprotection of the OH-groups are involved. Enzymatic glycosylation appears to be even easier because protection/deprotection is not necessary. Nevertheless, both chemical and enzymatic arbutin-syntheses continue to be published to the present day, some with novel twists. This review covers the most important developments.

## 2. Chemical Syntheses of Arbutin

The multi-step arbutin synthesis starting from glucose by C. Mannich in 1912 resulted in a poor overall yield, but for the first-time, care was taken to purify the compound [23]. Glucose was first treated with an excess of acetyl bromide in the absence of a solvent, resulting in the per-acylated form with Br at the anomeric center. This glycosyl bromide was then treated with HQ under basic conditions (NaOH), leading to tetra-acetyl arbutin. Finally, hydrolytic deprotection was achieved by treatment with a barium hydroxide solution (“Barytwasser”), followed by acidification using CO_2_ [23]. Subsequent reports appeared in which the Koenigs-Knorr reaction was applied, specifically on the basis of the Helferich variant utilizing phenol-type donors, which proved to be more efficient. A more recent study was published in 2004 that featured a multi-step process including the protection and deprotection of HO-groups in glucose [29]. In yet another study, rather than utilizing the glycosyl bromide, the respective acetate was employed, which offers additional advantages (Scheme 1) [30]. The glycosylation reaction was performed with the Lewis acid BF_3_·OEt_2_ (55–62% yield) followed by complete deprotection (92%) [30].

A successful variation of this theme, where the hydroxyl group of HQ is not protected, was subsequently reported in 2008 (Scheme 2) [31]. In this case, the same Lewis acid was applied, but instead of refluxing, microwave conditions were chosen.

Organic chemists may wonder why the glycosyl trichloroacetimidates, so successfully developed by R. R. Schmidt for many transformations [32], have not been used in the synthesis of arbutin. The reason is that a 2,3,4,6-tetra-benzyl protected trichloroacetimidate was shown to react with HQ under catalysis by trimethylsilyltriflate (TMSOTf) with the aformation of diastereomeric α-arbutin derivative [26], which is not the protected form of the natural product arbutin (β-arbutin). Several other selected key papers of unnatural α-arbutin are listed here [27,28].

## 3. Biosyntheses of Arbutin

A number of biosyntheses of arbutin have been published; only a few key papers are considered here. In systematic studies of the biosynthesis of plant glycosides, it was reported as early as 1960 that an enzyme from wheat germ catalyzes the reaction of uridine diphosphate glucose [UDPG,(**6**)] with HQ (**5**) to form arbutin (**1**) and uridine diphosphate [UDP, (**7**)] (Scheme 3) [2].

### 3.1. Biocatalytic Syntheses of Arbutin Using 4-Hydroxybenzoic Acid as the Starting Substrate

More recently, upon exploiting the shikimate pathway [33], an enhanced biosynthesis of arbutin was achieved [34]. A plasmid-free biosynthetic pathway was constructed in *Pseudomonas chlororaphis* P3, which is a mutant strain derived from *P. chlororaphis* HT66 following multiple rounds of chemical mutagenesis and selection. It has been widely used for phenazine production. The *P. chlororaphis* P3-Ar5 strain was constructed using the native promoter P_phz_ by way of chromosomal integration (Figure 3) [34]. Mixed fed-batch fermentation of glucose and 4-hydroxybenzoic acid (4-HBA) as a precursor enabled arbutin production of 6.79 g/L with a productivity of 0.094 g/L/h, which constitutes a 54-fold improvement relative to the starting strain. A different biosynthetic pathway for arbutin production starting from glycerol was also reported [34].

Other approaches especially include the use of sugar-nucleotide-dependent (Leloir-type) glycosyltransferases, which enable a wide range of synthetically important transformations in carbohydrate chemistry, as recently reviewed [35]. Leloir glycosyltransferases utilize activated sugar donors, typically sugar nucleotides, as substrates. In a 2019 review, the metabolic engineering of microorganisms for the production of structurally different aromatic compounds was critically analyzed [33]. This review summarized the latest metabolic engineering strategies and tools applied to the biosynthesis of aromatic chemicals, many derived from shikimate and aromatic amino acids including arbutin. For example, arbutin production was enhanced by engineering the shikimate pathway and optimizing the initial glucose concentrations. In principle, all of these methods can be applied for the production of arbutin.

### 3.2. Biocatalytic Syntheses of Arbutin Using Benzene as the Starting Substrate

Recently, a conceptionally different biocatalytic synthesis of arbutin was reported [36]. It is unusual because it does not require HQ as a starting material. The motivation for using benzene as the starting substrate has to do with the operational difficulty that chemists face when synthesizing HQ. The current industrial process for producing this compound involves several steps starting from benzene, which includes the formation of a potentially explosive di-alkylperoxide and acetone as a side product, which needs to be separated (Figure 4A). Alternative syntheses involving the transformation of benzene into phenol followed by hydroxylation poses problems due to overoxidation and lack of strict regioselectivity at the *para*-position [37]. Thus, we set out to devise a biocatalytic ecologically viable (green) route to HQ (Figure 4B), and to develop a cascade sequence stretching from benzene all the way to arbutin in a one-pot process (Figure 4B and Figure 5) [36].

The basic challenge was the transformation of benzene into HQ (Figure 4B), for which we considered the cytochrome P450 monooxygenase from *Bacillus megaterium* (P450-BM3), an enzyme that mediates the hydroxylation of fatty acids as natural substrates, and that has been engineered many times by rational design or directed evolution [38]. Unfortunately, it does not readily accept “small” compounds such as propane, cyclohexane, or benzene. Based on an earlier directed evolution study of P450-BM3 as a catalyst in the oxidative hydroxylation of small molecules, mutational “hotspots” have been identified [39]. On the basis of this information, we first semi-rationally designed and tested three mutants in the reaction of benzene, namely A82F, A82F/A328F, and V78F/A82F/A328F. In particular, the double and triple mutants proved to be excellent biocatalysts (Table 1). The double mutant A82F/A328F was then tested in upscaled reactions using whole *E. coli* cells, starting from either benzene or phenol as the substrates, which led to 55–76% isolated yields of pure HQ with essentially no undesired overoxidation [36].

We then focused on the one-pot cascade conversion of benzene to arbutin (Figure 5). In the quest to find an active glucosyltransferase for the selective mono-glycosylation of HQ, we turned to the UDP-glucose dependent glucosyltranserase from *Rauvolfia serpentina* as the arbutin synthase (AS). It had previously been demonstrated that it showed high activity for HQ and only 5% activity for the glucosylation of phenol [40,41]. This was ideal for our aim, because glucosylation of the intermediate phenol in our cascade sequence would have inhibited arbutin production. Therefore, this glucosyltransferase (AS) was cloned and expressed in *E. coli.* Upon testing whole cell activity for HQ acceptance, 95% conversion was observed for arbutin formation within 5 h. The designed whole cell system was tested for the entire sequence starting with benzene (5 mM). As anticipated, high yields of arbutin amounting to 80–83% were observed [36].

Finally, the remarkable regioselectivity deserves comment, because the transformation of mono-substituted benzene derivatives is known to be ortho-selective. In order to shed light on the mechanism of the oxidative hydroxylation of phenol, quantum mechanics/molecular mechanics (QM/MM) simulations were performed, suggesting that the C3–C4 double bond rather than the C2–C3 double bond of phenol becomes selectively epoxidized, and then the intermediate characterized by a C3–C4 epoxy function leaves the binding pocket and undergoes the expected rapid non-catalyzed H_2_O-assisted fragmentation [36].

### 3.3. Biocatalytic Syntheses of Arbutin Derivatives and α-Arbutin

Arbutin precursors in which some or all of the hydroxyl groups are protected can be considered to be arbutin derivatives with different biological properties. A different and useful concept involves the enzymatic regioselective acylation of arbutin itself. A prominent example is the acylation of arbutin, catalyzed by the lipase from *Candida antarctica* B (CALB) in organic solvents, with regioselective formation of feruloyl arbutin and lipoyl arbutin without the necessity to employ vinyl ferulate or vinyl lipoate, respectively [42]. Additionally, biocatalytically and chemically controlled polymerizations of arbutin have been reported a number of times, the novel properties of these polymers being of special interest, as in the case of a Cu-mediated process [43]. Finally, natural arbutin analogs also deserve attention because they open many new doors for a variety of different applications [44].

It is interesting to compare the biocatalytic steps to arbutin with biosynthetic routes to the unnatural diastereomeric α-arbutin [1,2,3,4,5,6,26,27,28]. For example, the amylosucrase encoding gene from *Deinococcus geothermalis* DSM 11300 was engineered and expressed in *E. coli*, leading to an efficient platform for constructing the biosynthesis of catechol glucosides, resorcinol glucosides, and hydroquinone glucosides (α-arbutin) from 1,2-, 1,3-, and 1,4-dihydroxybenzene (HQ), respectively [45]. The example relevant to α-arbutin production utilizes sucrose as the glucose donor source and HQ, which enables a molar yield of the product amounting to 325.6 mM (88.6 g/L).

## 4. Conclusions

The natural product arbutin, a glucoside derived from hydroquinone (HQ, 1,4-dihydroxybenzene) occurring in the leaves of certain plants, has a long history dating back to a report by A. Kawalier in 1852 [21]. Following a short historical account, this review covered the recent essential developments concerning its structure, biological properties, and in particular, the synthesis of arbutin, which can be performed by modern chemical methods or by biocatalytic processes. Until recently, all of them have been based on the use of hydroquinone (HQ, 1,4-dihydroxybenzene). Modern chemical syntheses utilize the Koenigs-Knorr glycosylation method, but require multi-steps due to the necessity of protecting and deprotecting the respective hydroxyl moieties. In contrast, the advantage of biosynthetic approaches concerns the fact that such protection/deprotection steps are not necessary. Nevertheless, they require the use HQ, which in turn is still being produced industrially by a multi-step process that is no longer considered to be a green technique.

The most recent one-pot biocatalytic production of arbutin circumvents the use of HQ because it starts with benzene as the feedstock. In this unusual approach, benzene is first chemo- and regioselectively dihydroxylated in a process that is catalyzed by a mutant of cytochrome P450-BM3 accessible by directed evolution. The in situ formation of HQ is followed by the required glucosylation, catalyzed by an appropriate glucosyltransferase. The one-pot whole cell cascade reaction sequence in *E. coli* starting from benzene leads to an 80–83% yield of arbutin. It is important to note that the selective whole cell dihydroxylation of benzene can be stopped at the stage of HQ. This means that the stage is set for using these designer cells in the biocatalytic production of other higher-value products based on benzene, also in one-pot cascade sequences.

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
