# Peer review of "Chemical and Biocatalytic Routes to Arbutin"

_molecules, 2019, doi:10.3390/molecules24183303_

Round 1

Reviewer 1 Report

This manuscript is a review about chemical and biochemical synthesis of b-arbutin.  Occurrence, importance, and history of the synthesis of arbutin including the authors work were compactly reviewd.  Though the reviewer is not a specialist of natural products and biosynthesis, the reviewer can understand the chemistry of arbutin.  Therefore, the reviewer considers that this manuscript is acceptable to publish in your Journal.

The reviewer points out some improvements.

Number of schemes were written in Roman numbers (Scheme I, Scheme II, etc), but they were written in Arabic numbers in the text (Scheme 1, Scheme 2, etc). Section 2 deals with chemical synthesis of arbutin, whereas section 3 and 4 deal with biochemical synthesis of arbutin: section 3: from HQ and 4-HBA, section 4: from benzene through HQ. The reviewer thinks that section 3 and section 4 would be section 3-1 and section 3-2, respectively. Figure 4, Eq. C. It would be more kind to readers if chemicals numbers for UDPG and UDP were shown (UDPG (6), UDP (7)).  B can be omitted because it is a part of Eq. C. In line 203, please indicate “QM/MM simulation”.

Reviewer 2 Report

In the manuscript entitled “Chemical and Biocatalytic Routes to Arbutin” which is submitted as a review format, the author mainly described the syntheses of arbutins which include chemical, biosynthetic, and biocatalytic methods.

Arbutin is a naturally occurring beta-D-glucopyranoside of hydroguinone (HQ) and it displays a wide range of pharmacological properties such anti-microbial, anti-oxidant, and anti-inflammatory activities. In particular, arbutin has been used worldwide for the treatment of skin ailments by way of depigmenting as a component of numerous products of the cosmetics and healthcare industries. In this review, the author summarized some important aspects of naturally occurring arbutin including a short historical account, isolation, biological properties, and syntheses with a particular emphasis on biocatalytic method which was based on the recent publication from author’s laboratory (Ref. [40], Angew. Chem. Int. Ed. 2019, 58, 764-768).

In contrast to Part 4 (4. Biocatalytic synthesis of arbutin using benzene as the starting substrate), Part 3 (Biosynthesis of arbutin) was not described in detail and it is hard to understand some part of the text. Therefore, this reviewer suggests that the authors should provide more description or some figures on the text of part 3 (in particular Ref. [35]-[40]).

In the Figure 3, the structure of glucose portion of both arbutin and its dimeric structure were wrong. The authors provided L-glucose which is a mirror image of D-glucose. Naturally occurring arbutin should have D-glucose, so the glucose structure in Figure 3 should be revised to D-glucose. Also, Figure 3 is a duplicate of Figure 1 in the Ref. [34]. This reviewer thinks that the authors should obtain permission from the authors of Ref. [34] or the publishing party on using Figure 3 in this review.

In summary, this review will be of interest to the general leadership of Molecules and this reviewer recommends the publication of this manuscript in the Journal, if the following punctuation or typographical errors found in the text are also suitably considered.

Page 03, Line 81; Revise “CO2” to CO2”. Page 03, Scheme 01; Revise “BF3·EtH2O” to “BF3·Et2O”. Page 03, Line 87; Revise “BF3-etherate” to “BF3·OEt2”. Page 03, Line 090; Revise “Scheme I (Roman)” to “Scheme 1 (Arabic)”. Page 03, Line 095; Revise “Scheme II (Roman)” to “Scheme 2 (Arabic)”. Page 04, Line 110; Revise “Scheme III (Roman)” to “Scheme 3 (Arabic)”. References; Some of references have only first page number. This reviewer suggests that final page number in those references should appear.

Round 2

Reviewer 2 Report

In this revised manuscript entitled "Chemical and Biocatalytic Routes to Arbutin" submitted as a review format, the authors have been responsive to the reviewer's comments from the first round of review and the manuscript is well-revised and has improved as a result.

However, in the Figure 3 of the revised manuscript, the structure of glucose presented by the authors was still incorrect. Naturally occurring arbutin has D-form of glucose, but both arbutin and its dimeric structure in the Figure 3 is L-form of glucose. In addition, the structures of pyranoses in the cover letter for the revised manuscript were incorrect. The left structure is “L-glucose” (not D-glucose), and the right structure is “D-idose (not L-glucose). D-idose is not an enantiomer of L-glucose, but an epimer (or diastereomer). This reviewer has also checked the structures of arbutin in the references 34 and 41 and found that they are also incorrect. Arbutin in both references has L-form of glucose.

Therefore, this reviewer recommends the publication of this manuscript in the Journal, if the authors carefully re-check the structures of glucoses in the Figure 3 and this issue is suitably solved.
